# Adeno-Associated Viral Transfer of Glyoxalase-1 Blunts Carbonyl and Oxidative Stresses in Hearts of Type 1 Diabetic Rats

**DOI:** 10.3390/antiox9070592

**Published:** 2020-07-06

**Authors:** Fadhel A. Alomar, Abdullah Al-Rubaish, Fahad Al-Muhanna, Amein K. Al-Ali, JoEllyn McMillan, Jaipaul Singh, Keshore R. Bidasee

**Affiliations:** 1Department of Pharmacology and Toxicology, College of Clinical Pharmacy, Imam Abdulrahman Bin Faisal University, Dammam 31441, Saudi Arabia; 2Department of Internal Medicine, College of Medicine, Imam Abdulrahman Bin Faisal University, Dammam 31441, Saudi Arabia; arubaish@iau.edu.sa (A.A.-R.); fmuhanna@iau.edu.sa (F.A.-M.); 3Institute for Research and Medical Consultation, Imam Abdulrahman bin Faisal University, Dammam 31441, Saudi Arabia; aalali@iau.edu.sa; 4Department of Pharmacology and Experimental Neuroscience, University of Nebraska Medical Center, Omaha, NE 68198-5800, USA; jmmcmillan@unmc.edu; 5Environmental, Agricultural and Occupational Health, University of Nebraska Medical Center, Omaha, NE 68198-5800, USA; 6College of Science and Technology, University of Central Lancashire, Preton PR1 2HE, England, UK; jsingh3@uclan.ac.uk; 7Nebraska Redox Biology Center, Lincoln, NE 68588-0662, USA

**Keywords:** diabetes mellitus, cardiomyopathy, methylglyoxal, glyoxalase-1, carbonyl stress, oxidative stress, adeno-associated virus

## Abstract

Accumulation of methylglyoxal (MG) arising from downregulation of its primary degrading enzyme glyoxalase-1 (Glo1) is an underlying cause of diabetic cardiomyopathy (DC). This study investigated if expressing Glo1 in rat hearts shortly after the onset of Type 1 diabetes mellitus (T1DM) would blunt the development of DC employing the streptozotocin-induced T1DM rat model, an adeno-associated virus containing Glo1 driven by the endothelin-1 promoter (AAV2/9-Endo-Glo1), echocardiography, video edge, confocal imaging, and biochemical/histopathological assays. After eight weeks of T1DM, rats developed DC characterized by a decreased E:A ratio, fractional shortening, and ejection fraction, and increased isovolumetric relaxation time, E: e’ ratio, and circumferential and longitudinal strains. Evoked Ca^2+^ transients and contractile kinetics were also impaired in ventricular myocytes. Hearts from eight weeks T1DM rats had lower Glo1 and GSH levels, elevated carbonyl/oxidative stress, microvascular leakage, inflammation, and fibrosis. A single injection of AAV2/9 Endo-Glo1 (1.7 × 10^12^ viron particles/kg) one week after onset of T1DM, potentiated GSH, and blunted MG accumulation, carbonyl/oxidative stress, microvascular leakage, inflammation, fibrosis, and impairments in cardiac and myocyte functions that develop after eight weeks of T1DM. These new data indicate that preventing Glo1 downregulation by administering AAV2/9-Endo-Glo1 to rats one week after the onset of T1DM, blunted the DC that develops after eight weeks of diabetes by attenuating carbonyl/oxidative stresses, microvascular leakage, inflammation, and fibrosis.

## 1. Introduction

Heart failure is a major cause of morbidity and mortality in patients with chronic diabetes mellitus [1]. In patients with Type 1 diabetes mellitus (T1DM), diabetic cardiomyopathy (DC) starts with an impairment in diastolic dysfunction and progresses to systolic dysfunction [2,3,4,5]. Increases in oxidative stress, inflammation, fibrosis, sympatho-excitation, activation of the renin-angiotensin system, impairment in myocyte intracellular Ca^2+^ handling, mitochondrial dysfunction, and changes in miRNA composition have been identified as contributing causes of both diastolic and systolic dysfunctions [2,3,4,5,6,7,8,9,10,11,12,13]. Data from several recent studies suggest that the accumulation of the toxic glycolytic metabolic methylglyoxal (MG) may be an elusive cue that is responsible for the diverse pathobiologies seen in DC [14,15,16,17,18].

In healthy cells, about 0.1% of the glucotriose flux is converted into MG during the interconversion of glyceraldehyde 3-phosphate and dihydroxyacetone phosphate [19]. Smaller amounts of MG are also produced from the spontaneous breakdown of glucose and lipids [20]. Free MG is rapidly degraded by the dual glyoxalase system to minimize cellular toxicity. Glyoxalase-1 (Glo1) is the rate-limiting enzyme and converts a hemiacetal formed between MG and glutathione (GSH) into S-D-lactoylglutathione [19,21]. In turn, Glo2 then hydrolyzes this thioester to D-lactate and GSH. Expression of Glo1 is regulated by the antioxidant transcription factor, nuclear factor erythroid 2-related factor 2 (Nrf-2) [19,22].

In an earlier study, we showed that addition of MG to freshly isolated ventricular myocytes perturbed intracellular Ca^2+^ homeostasis and increased mitochondrial reaction oxygen species (mROS) production [17]. These effects were due in part to the formation of irreversible MG adducts on critical basic residues of Ca^2+^ cycling and electron transport chain proteins needed for functioning [17,18,23]. We also showed that bathing pial arterioles on the top of rat brains (in vivo) with MG, attenuated endothelial-cell mediated vasodilatation but not smooth muscle-medicated vasodilatation of these arterioles [24]. We and others also identified an inversed relationship between endothelial cell function, microvasculature leakage, and MG [15,24]. Others have also shown that MG can activate the inflammation transcription factor nuclear factor kappa light chain enhancer of activated *B* cells (NF-κB) in other cell types [25].

Schalkwijk and colleagues were the first to report that transgenic rats overexpressing human Glo1 were less susceptible to T1DM-induced cardiac oxidative damage, inflammation, and fibrosis [14]. Suuronen and colleagues later showed that transgenic mice overexpressing human Glo1 under regulation of the preproendothelin-1 promoter were protected against T1DM-induced impairments in fractional shortening and ejection fraction, and that these protections were due in part to improvements in coronary microvascular endothelial function and attenuation in carbonyl stress [15]. Others have also reported that treatment with small molecule activators of Nrf2 to increase Glo1 expression also protected against DC [26]. Whether expressing Glo1, using gene transfer strategies after the onset of T1DM, would also blunt the development of DC has not yet been determined.

Adeno-associated viruses (AAV) are increasingly being used as safe and effective vectors for delivering genes to target cells for therapeutic gains [27,28,29]. Cardiac targeting can also be attained by using engineered AAV serotype with a select capsid and gene promoters [30,31]. Since inflammation is upregulated during T1DM, we created an AAV serotype 2 with capsid 9 driven by the promoter of the inflammation-regulated protein, endothelin-1 (AAV2/9-Endo-Glo1) [32] to (1) determine if expressing Glo1 in the hearts of rats, shortly after the onset of T1DM, would be cardio-protective, and (2) to compare tissue, cellular, and molecular features of hearts from Glo1-treated and untreated T1DM hearts to better understand the mechanisms that contribute to the development of DC.

## 2. Materials and Methods

### 2.1. Antibodies and Reagents

Goat polyclonal anti-TAGLN [SM22α] antibodies were from AbCam Inc. (Cambridge MA, USA); mouse monoclonal, MG-H1, [1H7G5] antibodies were from Hycult Biotech (Wayne PA, USA); rabbit polyclonal Glo1 [FL-184] antibodies, actin [1,2,3,4,5,6,7,8,9,10,11,12,13,14,15,16,17,18,19] antibodies, donkey anti rabbit IgG-HRP, and donkey anti goat IgG-HRP were from Santa Cruz Biotechnology Inc. (Santa Cruz, CA, USA); NF-κB p65 and phosphor-p65 (Ser536, NF-κB) were from Cell Signaling Technologies (Danves MA, USA); chicken anti-rabbit IgG coupled to Alexa Fluor 488, chicken-anti-mouse IgG coupled to Alexa Fluor 488, and donkey anti-goat IgG coupled to Alexa 594 were obtained from Invitrogen Life Technologies (Carlsbad, CA, USA); Fluoroshield with DAP1, bovine serum albumin labeled with fluorescein isothiocyanate (BSA-FITC), and the Trichrome (Masson) staining kit (Cat# HT15-1KT) were from Sigma-Aldrich, St Louis MO. Primers (TNF-α, sense, ATGAGCACTGAAAGCATGAT and antisense, CTCTTGATGGCAGAGAGGAG and β-actin, sense, CGTAAAGACCTCTATGCCA and antisense AGCCATGCCAAATGTCTCAT) were from Integrated DNA Technologies (Coralville, IA, USA). The reduced and oxidized glutathione assay kit was from Oxis Research (Portland, OR, USA). The total thiobarbituric acid reactive substances (TBARS) assay kit was from (Zepto Metric Corporation, Buffalo, NY, USA). All other reagents were from commercial sources.

### 2.2. Construction of AAV2/9 Containing Glo1 and eGFP Driven by the Endothelin 1 Promoter

The University of Pennsylvania Vector Core Facility with assistance from the Gene Therapy Resource Program (GTRP#1053), generated the adeno-associated viruses containing rat glyoxalase-1 (Glo1) and eGFP driven by the endothelin-1 promoter (AAV2/9-Endo-Glo1 and AAV2/9-Endo-eGFP) that were used in this study [24].

### 2.3. Induction, Verification, and Treatment of T1DM Rats

Animal procedures were approved by the Institutional Animal Care and Use Committee, University of Nebraska Medical Center (IACUC #02-077-11 and #15-117-12-FC). Seven-weeks old male Sprague-Dawley rats were administered a single intravenous injection of streptozotocin (STZ, 40–45 mg/kg, in 0.1 mL in citrate buffer pH 4.5) [17,18,24] to induce T1DM. Seven days after STZ injection, blood glucose levels were measured using an Accu Chek glucose stick (Roche Molecular Biochemicals, Indianapolis, IN) and animals with blood glucose >300 mg/dL (16.7 mmol) were divided into three groups. The animals in Group 1 were administered a single intravenous injection of AAV2/9-Endo-Glo1 (1.7 × 10^12^ viron particles/kg in sterile physiologic saline solution, T1DM-Endo-Glo1). The animals in Group 2 were given a single injection of AAV2/9-Endo-eGFP (1.7 × 10^12^ viron particles/kg) while the animals in Group 3 remained untreated for the eight week protocol (T1DM). Control animals injected with citrate buffer only were also divided into two groups. One group was injected with AAV2/9-Endo-eGFP, and the other group remained untreated (Con). The dose of virus used in this study was as per our recent study [24] and agrees with other studies [30,31].

### 2.4. Assessment of Left Ventricular Function

Echocardiography was performed using a Vevo 2100 Echocardiograph system (Fujifilm VisualSonics, Toronto, ON, CAN) employing a MX201 transducer with a center frequency of 15 Hz and an axial resolution of 100 µM to assess left ventricular function in rats prior to and eight weeks after injection of STZ or saline. For this, hair on chests of rats was removed (Nair, Church & Dwight Co., Inc. New Jersey, USA) and anesthesia was induced with 3% isoflurane (Cardinal Health, Dublin OH, USA). Anesthesia was maintained with 1–2% isoflurane for assessment of the in vivo cardiac function.

(a) Pulsed-wave Doppler in the parasternal short axis mode was used to acquire diastolic parameters. The offline Program Vevo LAB 3.1.1 was used to measure peak early- and late-diastolic transmitral velocities (E and A waves) and isovolumetric relaxation time (IVRT). E:A ratios were also calculated.

(b) Tissue Doppler was used to assess early-diastolic tissue relaxation velocity (e’) and to calculate the E/e′ ratio.

(c) M-mode Doppler in the parasternal long axis view was used to assess left ventricular (LV) mass, fractional shortening (FS), and percent ejection fraction (EF), left ventricular end diastolic diameter (LVEDD), and left ventricular end systolic diameter (LVESD).

(d) Speckle tracking analysis of parasternal long axis B-mode images obtained at a rate of >300 frames/second were also in the “reverse peak” mode to assess early ventricular stiffness [33].

### 2.5. Assessment of Myocyte Function

Ventricular myocytes were isolated as described earlier [17,18]. After isolation, myocytes were resuspended in Dulbecco’ modified eagle medium (DMEM) containing 1.8 mM Ca^2+^, supplemented with F-12 (1:1) and antibiotics (100 U/mL penicillin, 100 μg/mL streptomycin, and 100 μg/mL gentamicin). All functional measurements were made within 4 h after isolation.

(a) Evoked contractile kinetics: Ventricular myocytes in DMEM were electrically stimulated (0.5 Hz, 10 V, and 10 ms) using platinum wires and contraction/relaxation were recorded using IonOptix Data Acquisition Software (Corporation, Milton, MA). Rates of shortening, lengthening, and extent of cell shortening were determined using IonWizard, Version 5.0 [17,18].

(b) Evoked Ca^2+^ transients: Ventricular myocytes were loaded with Fluo-3 AM (5 μM) as described earlier [17,18]. Cells were field-stimulated (0.5 Hz, 10 V, and 10 ms) and evoked Ca^2+^ transients were recorded using a 510 Zeiss confocal microscope. Fluo-3 was excited by light at 488 nm, and fluorescence emission was measured at 525 nm. LSM Meta 5.0, Prism 5.0, and Microsoft Excel were used for data analyses.

### 2.6. Histopathological Analyses

(a) Vascular perfusion and permeability: Rats were injected intraperitoneally with heparin (100 unit/kg, i.p.). Five min later, they were anaesthetized with 3% isoflurane and injected via a tail vein with bovine serum albumin coupled to fluorescein isothiocyanate (BSA-FITC, 50 mg/kg in sterile PBS buffer) [24,34]. Chest cavities were opened five minutes after BSA-FITC, and hearts were removed and fixed in 4% paraformaldehyde overnight at 4 °C. Hearts were transferred to 4% paraformaldehyde/15% sucrose solution for 24 h, 4% paraformaldehyde/30% sucrose solution for 24 h. and then 30% sucrose solution for 24 h before embedding in optimal temperature cutting (OTC) medium and storing at −80 °C until use. Cryoprotected hearts were cut longitudinally into 20 μm sections (Leica EM-UC 6 cyostat, Leica Microsystems, Wien, Austria) and mounted onto pre-cleaned glass slides. Sections were washed three times with 1X PBS and cover slipped with Vectashield^®^ mounting medium with DAPI. A Nikon inverted fluorescence microscope (TE 2000) equipped with a CoolSNAP HQ2 CCD Camera (Photometrics, Tucson, AZ) and 10× lens was used to visualize the permeability of BSA-FITC as indices of microvascular leakage and ischemia. The density of micro-vessels (length ≥40 µm) perfused with BSA-FITC per 10× frame was used as an index of micro-vessel perfusion. Microvascular leakage was determined by the number of BSA-FITC was a “halo or blob” per 10× frame.

(b) Fibrosis: Hearts fixed in 4% paraformaldehyde overnight at 24 °C were embedded in paraffin as described earlier [34]. Five micrometer coronal ventricular sections were then cut and placed on glass slides, de-paraffinized, rehydrated and fibrosis was assessed using the Masson Trichrome staining kit as per the manufacturer’s instructions. A Zeiss ApoTome inverted microscope was used to assess fibrotic staining with a 10× lens.

(c) Immunofluorescence: Immuno-fluorescence assays were performed on 5 µm paraffin sections to determine ventricular levels of MG-H1 (MG-hydro-imidazolone isomer 1) and Glo1 as described earlier [24]. Calponin-related protein (SM22α), a marker of contractile smooth muscle cells served as an internal reference. Primary antibodies were used at 1:100 dilutions and secondary antibodies were at 1:250 dilutions. Horse serum (10%) was used to reduce non-specific interactions. Images were taken with a Nikon inverted fluorescence microscope (TE 2000) at 40× equipped with a CoolSNAP HQ2 CCD Camera.

### 2.7. Carbonyl Stress, Oxidative Stress, and Inflammation

(a) Carbonyl stress: MG-H1 on cardiac proteins was also determined using Western blot assays [17,18,35,36]. Primary antibody concentrations were 1:1000 for 16 hrs at 4 °C and secondary antibody concentrations were 1:2000 for 2 hrs at room temperature. β-actin served as a control for variations in sample loading. Protein bands with increased MG-H1 immunoreactivity were excised and their identities were determined using mass spectrometry (Quadruple time-of-flight mass spectrometry, Micromass, Manchester, UK).

(b) Oxidative stress/reactive oxygen species (ROS): Freshly isolated ventricular myocytes were loaded with MitoTracker Green (100 nM) and MitoSOX™ (2 μM, a marker for ROS) [36]. Cells were then excited at 488 nm and emissions at 594 nm (MitoSOX™ Red) and 516 nm (MitoTracker Green) and they were measured using a Zeiss LSM 510 Meta laser scanning microscope, to assess ROS in mitochondria. Images were quantified with ImageJ analysis software [37].

(c) Inflammation: Phosphorylated levels of the p65 (Ser536) subunit of nuclear factor–kappa light-chain-enhancer of activated B cells (NF-κB) and tumor necrosis factor alpha TNF-α mRNA were used as indices of inflammation as described earlier [24].

### 2.8. Glo1 in Ventricular Homogenates

Ventricular homogenates (500 μg) were prepared as described earlier, and Glo1 activity was assayed by measuring the rate of formation of S-D-lactoylglutathione from hemi-thioacetal [24]. A standard curve was generated using commercially available Glo1 standard and used for quantification. Glo1 levels in ventricular homogenates were determined using Western blot assays as described earlier to correlate with activities. Primary antibody concentrations were 1:1000 for 16 h at 4 °C and secondary antibody concentrations were 1:2000 for 2 h at room temperature. β-actin served as the internal control to correct for variations in sample loading.

### 2.9. Methylglyoxal in Sera and Ventricular Homogenates

Methylglyoxal (MG) levels in sera and ventricular tissues were determined using the derivatization protocol originally described by Nemet et al., and modified and used by several labs [24,38,39,40]. A calibration curve was generated using commercially available 2-methylquinoxaline ((0.5–10 µM. cat #W511609, Sigma-Aldrich, St Louis MO, USA) for quantitation. In some samples (control and T1DM), o-phenylenediamine was left out during derivatization and in other, 5 µM 2-methylquinoxaline was spiked in to validate the retention time of 2-methylquinoxaline.

### 2.10. Total Thiobarbituric Acid in Sera and Ventricular Homogenates

Maliondialdehyde (MDA), another reactive aldehyde produced from oxidation of fatty acids in mitochondria, cytoplasm, and on the membranes of myocytes. Total thiobarbituric acid reactive substances (primarily) were measure in sera and ventricular homogenates using OXI-TEK (TBARS) assay kits (Zepto Metric Corporation, Buffalo, NY, USA) as per the manufacture’s instruction.

### 2.11. Glutathione Levels and Enzymes that Regulate GSH Production

Ratios of reduced to oxidized glutathione were determined in ventricular homogenates using a commercial kit according to the manufacturer’s protocol (Oxis Research, Portland, OR). γ-Glutamylcysteine ligase and glutathione reductase activities were determined using previously described procedures [24,41,42].

### 2.12. Statistical Analysis

One-way analysis of variance (ANOVA) followed by the Bonferroni’s post-hoc test was used for data analysis employing GraphPad Prism 7.0 (La Jolla, CA, USA). Data are presented in text as the mean ±S.E.M. Significance was determined at the 95% confidence interval with *p* < 0.05.

## 3. Results

### 3.1. General Characteristic of Animals

The general characteristics of the animals used in the study are summarized in Table 1. One week after STZ injection, blood glucose levels were >300 mg/dL and remained >300 mg/dL for the duration of the study. Body mass and plasma insulin levels of T1DM rats were lower and the mean % glycated hemoglobin was higher than that of controls at the end of the eight-week study. In prior studies, we showed that daily administration of insulin for two weeks starting six weeks after STZ injection, reversed increases in blood glucose, % glycated hemoglobin and loss in body mass indicating that these changes were not due to STZ toxicity [43,44]. Administration of AAV2/9-Endo-Glo1 one week after STZ injection, blunted reductions in body mass but not glucose or % glycated hemoglobin levels. Administration of the non-specific AAV2/9-Endo-eGFP one week after STZ injection, did not blunt body mass loss or attenuated blood glucose, plasma insulin, and % glycated hemoglobin. Injecting control rats with AAV2/9-Endo-eGFP had no impact on blood glucose, % glycated hemoglobin, body mass, plasma insulin, or MG levels (Table 1).

### 3.2. In Vivo Left Ventricular Function

(a) Conventional echocardiography: There were no significant changes in either diastolic or systolic functions one week after STZ-injection (data not shown). However, after eight weeks of T1DM, rats developed a DC characterized by grade 1 diastolic dysfunction with the E:A ratio decreasing from 1.46 to 1.15 (Figure 1A; see Table below) and E:e’ ratio increasing from 18.2 to 31.5. Isovolumetric relaxation time also increased from 13.5 to 20.5 ms. Percent fractional shortening and ejection fraction also decreased by 10% and 8.5% respectively (Figure 1B; see Table below). A single bolus intravenous injection of AAV2/9-Endo-Glo1 one week after STZ, blunted diastolic and systolic dysfunctions that developed eight weeks post STZ injection. Intravenous administration of the non-specific AAV2/9-Endo-eGFP one week after STZ injection did not blunt deleterious changes in myocardial function that were seen at the end of eight weeks of T1DM (Figure 1A,B).

(b) Global strain: speckle tracking (ST) analyses using the reverse peak algorithm revealed a 33% increase (*p* < 0.05) in longitudinal strain (peak, %) and 35% decrease in radial strain rates (peak, 1/s) eight weeks after STZ infection, which was in agreement with diastolic dysfunction (Figure 2A,B). Intravenous administration of AAV2/9-Endo-Glo1, but not AAV2/9-eGFP, one week after STZ injection, blunted the decrease in reverse longitudinal strain radial strain rates seen at the end of eight weeks T1DM.

### 3.3. Myocyte Function

(a) Evoked contractile kinetics: Eight weeks after STZ injection, evoked myocyte contraction and lengthening velocities were reduced by 33% (*p* < 0.05) compared to the controls (Figure 3, Table below). The extent of myocyte shortening was also reduced 25%. Intravenous administration of AAV2/9-Endo-Glo1 but not AAV/29-Endo-eGFP one week after STZ injection blunted impairments in myocyte contractile kinetics and cell shortening seen at eight weeks. Injecting control rats with AAV2/9-Endo-eGFP did not negatively impact myocyte contractile kinetics and evoked myocyte shortening.

(b) Evoked intracellular Ca^2+^ transients: Eight weeks after STZ injection, the rate of rise and amplitude of evoked myocyte Ca^2+^ transients decreased by 35% and 20%, respectively (*p* < 0.05) compared to the controls (see Figure 4 and Table below). Mean Ca^2+^ transient decay time was also doubled in the ventricular eight weeks after STZ injection (see Figure 4, Table below). Injecting rats with AAV2/9-Endo-Glo1, one week after STZ administration, blunted impairments in myocytes evoked Ca^2+^ transient kinetics seen at eight weeks after STZ injection. Injecting control and T1DM rats with AAV2/9-Endo-eGFP did not have any effect on evoked Ca^2+^ transient kinetics.

### 3.4. Histopathological Analyses

(a) Vascular perfusion and permeability: In control animals, the green fluorescence of BSA-FITC was seen throughout the coronary vascular network. Larger vessels exhibited higher BSA-FITC fluorescence compared to the smaller vessels (see Figure 5A, upper rows, inset white arrows), consistent with more BSA-FITC within. Injecting control rats with AAV2/9-eGFP had minimal impact of the pattern on BSA-FITC distribution in the heart (10× frames, see Figure 5A). In hearts from eight weeks T1DM rats, the pattern of BSA-FITC fluorescence was significantly different from that of control animals. At a first glance, the BSA-FITC was seen as “blobs or halos” and not confined to a vascular network as that in hearts from control animals. We interpreted this to mean that the BSA-FITC is being leaked into the myocardium. There were also several regions within the 10× frame of the myocardium from T1DM rats where there were little or no BSA-FITC fluorescence, suggestive of reduced perfusion and/or micro-ischemia (see Figure 5A, middle rows; white arrows). Hearts from rats injected with AAV2/9-Endo-Glo1 but not with AAV2/9-Endo-eGFP showed fewer BSA-FITC “blobs” seen per 10× frame, and a pattern that suggest BSA-FITC was confined within the vasculature (Figure 5A, lower pictograms). Quantitation of the density of vessels with BSA-FITC per 10× frame, and the number of BSA-FITC blobs per 10× frames are shown on graphs below (see Figure 5B,C).

(b) Myocardial fibrosis: eight weeks after STZ injection, myocardial tissues from T1DM rats show extensive blue Masson’s Trichrome staining consistent with fibrosis (see Figure 5D, middle panels, blue staining; yellow arrows). The increase in Masson’s Trichrome staining and fibrosis were seen both in the perivascular and interstitial space. Negligible blue Masson’s Trichrome staining was seen in myocardial tissues from the control animals (see Figure 5D, upper left panel). Administration of AAV2/9-Endo-Glo1 to rats one week after injection of STZ, blunted the increase in interstitial and perivascular fibrosis seen after eight weeks of chronic T1DM (Figure 5D). Quantification of data is shown below in the figure. Administration of the non-specific AAV2/9-Endo-eGFP to rats one week after STZ injection did not attenuate interstitial or perivascular fibrosis (Figure 5E).

(c) Immuno-fluorescence: eight weeks after STZ injection, Glo1 was reduced by 50% and 60% in cardiac myocytes and coronary microvascular smooth muscle cells (cSMCs), respectively compared to the controls (see Figure 6A–C). Immuno-reactive SM22-α was also lower in cSMCs of T1DM rats, suggestive of a reduction in cSMCs with a contractile phenotype. Administration of AAV2/-Endo-eGFP to rats one week after STZ injection did not blunt the loss of Glo1 in cSMCs (see Figure 6A, lower panel, and Figure 66C). However, there was a diffused green fluorescence within the myocytes, suggestive of eGFP expression (see Figure 6A, white arrows). Administration of AAV2/-Endo-Glo-1 to rats, one week after STZ injection, blunted the loss of Glo1 in myocytes and cSMCs (see Figure 6A, lower panel, and Figure 6B,C).

To validate changes in Glo1 seen in immuno-fluorescence studies, Glo1 protein and activity were also measured in ventricular homogenates. Eight weeks after STZ injection, the mean Glo1 activity in ventricular homogenates was 50% lower than that in the control animals (Table 1). This reduction in Glo1 activity arose primarily from a 50% reduction in Glo1 protein in ventricular homogenates (Figure 6D). A single intravenous injection of AAV2/-Endo-Glo1, but not AAV2/-Endo-eGFP to rats one week after STZ injection, blunted the reductions in ventricular Glo1 activity and protein levels (see Table 1 and Figure 6D). Injecting control rats with AAV2/-Endo-eGFP had no effects on ventricular Glo1 activity or protein expression (Table 1 and Figure 6D).

Eight weeks after STZ injection, MG-H1 levels in myocytes and cSMCs increased by 4.3- and 3.5-fold in cSMCs and myocytes respectively, compared to controls (see Figure 7A, 2nd panel white arrows, and 7B, 7C). Administration of AAV2/-Endo-eGFP to rats one week after STZ injection did not prevent MG-H1 formation in cSMCs and myocytes. Myocytes from AAV2/-Endo-eGFP-treated T1DM rats also exhibited a diffused green “auto-fluorescence” consistent with eGFP expression (see Figure 7A, 3rd panel red arrow). Administration of AAV2/-Endo-Glo-1 to rats, one week after STZ injection, blunted MG-H1 formation in myocytes and cSMCs (see Figure 7A, lower panel, and Figure 7B,C).

Western blots and mass spectrometry were also conducted to identify cardiac proteins with elevated MG-H1 adduct. After eight weeks of T1DM, the MG-H1 adduct was elevated on several Ca^2+^ cycling proteins including the ryanodine receptor (RyR2, 2.8 fold); myosin heavy chain beta (MHC-α/β, 2.9 fold); Na^+:^K^+^-ATPase (2.1 fold; sarco(endo)plasmic reticulum Ca^2+^ ATPase (SERCA2) (2.0 fold); potassium channel, K_v_ 4.2/4.3 (3.2 fold), and on mitochondrial NADH dehydrogenases (3.4 fold on NDUFV3 and 3.3 fold on NDUFV10; Figure 7E). Figure 7D also shows the Coomassie-stained SDS-PAGE gel that was used for the Western blot assay to indicate that the increases in MG-H1 adduct on these proteins were not from variations in the amount of proteins loaded per lane. Administration of AAV2/-Endo-Glo1 to rats one week after STZ injection blunted the increase in MG-H1 adducts on RyR2, SERCA2, MHC-α/β, Na^+^/K^+^-ATPase, and K_v_ 4.2/4.3 (see Figure 7F for RyR2 quantitation). We did not detect any significant reduction in the MG-H1 adduct on the mitochondrial NDUF proteins with AAV2/9-Endo-Glo1 treatment.

### 3.5. Oxidative Stress and Inflammation

(a) ROS levels in the ventricular myocytes: basal mitochondria ROS levels in cardiac myocytes isolated from chronic T1DM rat were 4× higher than that from non-diabetic rats (Figure 8A). In this study, we did not assess cytoplasmic ROS levels, but it is likely to be the same as mitochondria ROS. Administering AAV2/-Endo-Glo1 to rats one week after STZ injection blunted the increase in myocyte mitochondria ROS see in hearts untreated chronic T1DM rats (see Figure 8A,B).

Eight weeks after injection of STZ, TBARS (primarily MDA) levels were also elevated 4.2-fold and 4.0-fold in sera and ventricular homogenates, respectively compared to the controls (Table 1). A single intravenous injection of AAV2/-Endo-Glo1 but not AAV2/-Endo-eGFP to rats one week after STZ injection, significantly attenuated ventricular but not sera TBARS.

(b) TNF-α and NF-κB activation: mRNA levels for the pro-inflammatory mediator TNF-α and phosphorylation of the p65 protein of NF-κB were increased to 200% and 275% in the hearts rats with chronic T1DM, consistent with an increase in inflammation. A single intravenous injection of AAV2/-Endo-Glo1 but not AAV2/-Endo-eGFP to rats one week after STZ injection blunted the increase in TNF-α mRNA and p65 phosphorylation (see Figure 8C,D).

### 3.6. MG in Ventricular Sera and Homogenates

After eight weeks of T1DM, sera and ventricular MG levels were approximately four times higher than that in control animals (Table 1). A single intravenous injection of AAV2/-Endo-Glo1 but not AAV2/-Endo-eGFP to rats one week after STZ injection attenuated the increase in MG seen after eight weeks. Injecting control rats with AAV2/-Endo-eGFP did not alter sera and ventricular MG levels (Table 1).

### 3.7. Cellular GSH and GSH Regulatory Enzymes Levels

Eight weeks after STZ injection, the mean GSH:GSSG ratio was significantly lower (*p* < 0.05) in cardiac homogenates (Table 2) compared to controls. A single intravenous injection of AAV2/-Endo-Glo1 to rats one week after injection of STZ, blunted the reduction in GSH:GSSG. Injected rats with the non-specific AAV2/9-eGFP to rats one week after injection of STZ did not blunt the reduction GSH:GSSG ratio seen eight weeks after STZ injection.

γ-Glutamylcysteine ligase and glutathione reductase activities were also decreased by 55% and 32%, respectively in rat ventricular tissues eight weeks after injection of STZ compared to controls. A single intravenous injection of AAV2/9-Endo-Glo1 to rats one week after injection of STZ blunted the loss of γ-glutamylcysteine ligase and glutathione reductase activities. Administration of AAV2/9-Endo-eGFP to rats on the week after STZ injection did not increase the activities of γ-glutamylcysteine ligase and glutathione reductase.

## 4. Discussion

Although rates of T1DM-related cardiac complications have declined substantially over the past two decades, they are still >3 times higher than that of the general population [45]. These data emphasize the continuing need to better understand the molecular mechanisms that contribute to DC to identify novel strategies to attenuate them.

The principal finding of the present study was that expressing Glo1 in the hearts of rats one week after the onset of T1DM using adeno-associated virus driven by the promoter of the inflammation-regulated protein, endothelin-1, blunted the DC that develops after eight weeks of T1DM. This conclusion was based on several new observations. First, using in vivo echocardiography, we showed that expressing Glo1 in the hearts of rats one week after the onset of T1DM attenuated the diastolic and systolic dysfunctions that developed after eight weeks of chronic T1DM. Second, using video edge and live-cell confocal imaging, we showed that administering AAV2/9-Endo-Glo1 to rats a week after the onset of T1DM, attenuated impairments in myocyte contraction/relaxation kinetics and evoke Ca^2+^ transients that developed after eight weeks of chronic T1DM. Third, using biochemical, histopathologic and molecular approaches, we also showed that administration of AAV2/9-Endo-Glo1 to rats one week after the onset of T1DM attenuated cardiac microvascular leakage, fibrosis, inflammation/oxidative stress, formation of MG-HI adduct on Ca^2+^ cycling and mitochondrial proteins, and reductions in activities of γ-glutamylcysteine ligase and glutathione reductase and GSH levels. Since endothelin-1 is upregulated with inflammation and inflammation is increased in T1DM, we designed our virus to be driven by the endothelin-1 promoter to Glo1 during inflammation [32]. This strategy allowed Glo1 expression in the hearts of T1DM rats to be about 25% higher than that seen in non-diabetic animals.

Since Glo1 degrades the hemiacetal formed between MG and GSH (i.e., MG-GSH) [20], these data implicate, not only accumulation of MG, but also a reduction in GSH as key contributors of DC. GSH is synthesized by two ATP-dependent sequential reactions, ligation of l-glutamate and l-cysteine by γ-glutamylcysteine ligase (GCL; EC 6.3.2.2), and the addition of glycine to γ-glutamylcysteine by glutathione synthetase (GSS; EC 6.3.2.3) [46]. Glutathione reductase (EC 1.8.1.7) also known as glutathione-disulfide reductase (GSR) catalyzes the reduction of oxidized glutathione (GSSG) to the reduced glutathione (GSH), which is critical for maintaining the reducing environment inside cells. We earlier reported reduced GSH levels in diabetic myocytes [47]. In this study, we showed that the activities of both γ-glutamylcysteine ligase and glutathione reductase are reduced in rat hearts after eight weeks of T1DM. A reduction in the GSH pool will lead to a reduction in the formation of S-D-lactoylglutathione, resulting in “free MG”. Since MG is a potent inducer of ROS in the mitochondria of myocytes [17], the MG-induced increase in ROS will further deplete the GSH pool [48]. Thus, expressing Glo1 is likely to preserve the GSH level via multiple mechanisms. By converting available MG-GSH hemiacetal into the thioester (S-D-lactoylglutathione) for degradation by Glo2, it will replenish the GSH pool. Second, lowering MG will upregulate the ARE-Nrf2 pathway and the expression of γ-glutamylcysteine ligase and glutathione reductase, leading to an increase in the synthesis of new GSH [49].

In this study, we also showed that expressing Glo1 one week after the onset of T1DM, blunted the formation of the MG-H1 adduct on Ca^2+^ cycling (RyR2, SERCA2, MHC-α/β, Na^+^, K^+^-ATPase, and K_v_ 4.2/4.3) and mitochondria (NDUFV3 and NDUFV10) proteins. When the Glo1 and GSH level are reduced inside myocytes, the rate of formation and the amount of MG-GSH hemiacetal will be attenuated. The accumulated MG will instead react with susceptible basic amino acid residues on proteins including Ca^2+^ cycling (RyR2, SERCA2, MHC-α/β, Na^+:^/K^+^-ATPase, and K_v_ 4.2/4.3) and mitochondria (NDUFV3 and NDUFV10) proteins to form adducts. MG adducts that form on cysteine residues are usually reversible while those formed on lysine and arginine residues are irreversible [50]. Although MG-H1 was the only adduct assayed in this study, other MG adducts including N_ε_-(1-carboxyethyl)lysine (CEL) and argpyrimidine are also formed on RyR2 and SERCA2 during T1DM [17,18]. Others have shown that MG can also impair the function of key electron transport proteins [51,52]. Whether formation of the MG-H1 adduct on NDUFV3 and NDUFV10 also compromises their functions remains to be determined.

We also showed that expressing Glo1 in the hearts of rats shortly after injection of STZ, attenuated coronary microvascular leakage, impairment in vascular permeability, and fibrosis. When Glo1 protein is reduced as is the case in T1DM hearts, elevation in MG (either within the coronary circulation or diffusion from juxtaposed cells) can impair nitric oxide-mediated vasodilation of endothelial cells as well as reduce expression of the tight junction protein resulting in coronary microvascular leakage and microbleeds [24]. Microvascular leakage could also result in downstream ischemia. A reduction in nutrients to the myocytes will compromise pyruvate production and ATP needed for the activity of ATP-dependent proteins including ATPases. Microvascular leakage will also promote transcytosis of immune cells and other substances from the blood into the myocardium to activate matrix metalloproteinases (MMPs) to increase deposition of collagen fibers and ventricular stiffness [53], two key mediators of diastolic, and systolic dysfunctions. Since expressing Glo1 in the hearts of T1DM rats attenuated MG accumulation, it was not surprising that it also blunted NF-κB (p-65 subunit) activation and tumor necrosis factor alpha (TNF-α), since MG is an activator of NF-κB [25]. Whether expressing Glo1 in the hearts of T1DM rats also blunted other inflammation-induced pathways including activation of the protein-1 (AP-1) transcription factor [19] and the inflammasome remains to be determined.

## 5. Conclusions

In conclusion, the present study shows for the first time that administration of the adeno-associated virus containing glyoxalase-1 driven by the endothelin-1 promoter (AAV2/9-Endo-Glo1) to rats one week after onset of T1DM, attenuated the cardiomyopathy that develops after eight weeks of diabetes. The cardio-protective effect of Glo1 expression was independent of glucose lowering but associated with attenuation of known mediators of DC including carbonyl and oxidative stresses, microvascular leakage, ischemia, inflammation, fibrosis, and post-translational modification of proteins. More work is needed to determine if expressing Glo1 in female T1DM rats and in models of Type 2 DM would also be cardio-protective.

## Figures and Tables

**Figure 1 antioxidants-09-00592-f001:**
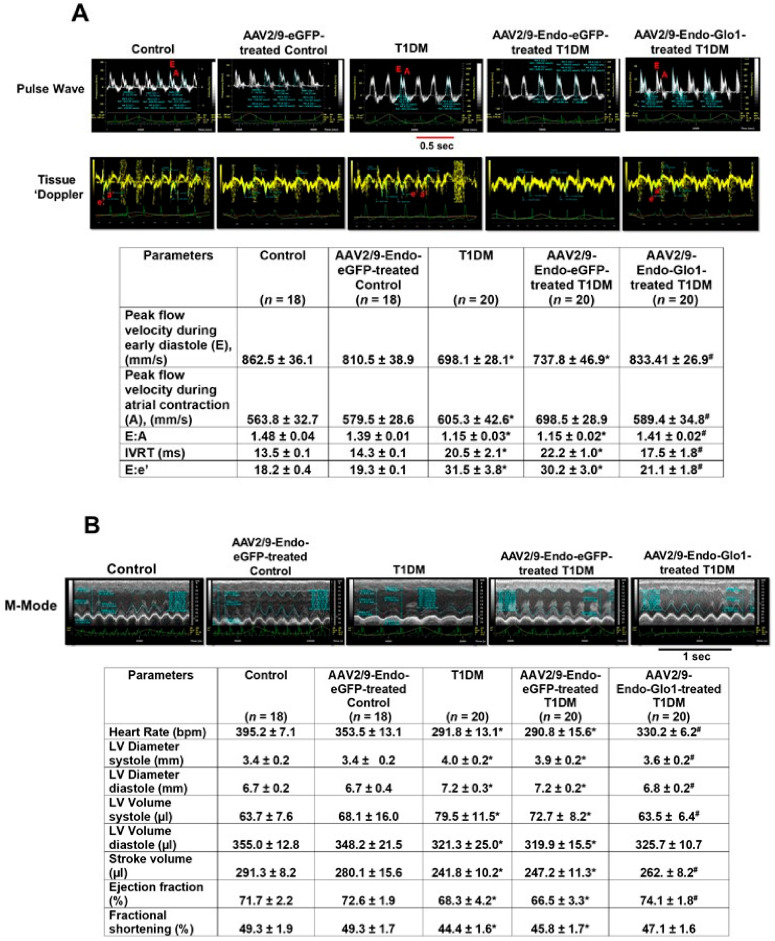
Original traces showing in vivo cardiac function of hearts from control, T1DM and Table 1. rats. (**A**) shows representative pulse-wave and tissue Doppler echocardiograms of hearts from control, AAV2/9-Endo-eGFP-treated Con, T1DM, AAV2/9-Endo-eGFP-treated T1DM, and AAV2/9-Endo-Glo1-treated DM rats. The table below shows mean ± S.E.M for peak flow velocity during early diastole (E), peak flow velocity during atrial contraction (A), E: A ratio, IVRT, and E:e’. (**B**) shows representative M-mode echocardiograms from control, AAV2/9-Endo-eGFP-treated Con, T1DM, AAV2/9-Endo-eGFP-treated T1DM, and AAV2/9-Endo-Glo1-treated DM rats. Hemodynamic data shown in Table below are mean ±S.E.M. Data shown are mean ± SEM. * denote significantly different from control (*p* < 0.05), # denote significantly different from T1DM (*p* < 0.05). The number of animals used per group is shown in the tables.

**Figure 2 antioxidants-09-00592-f002:**
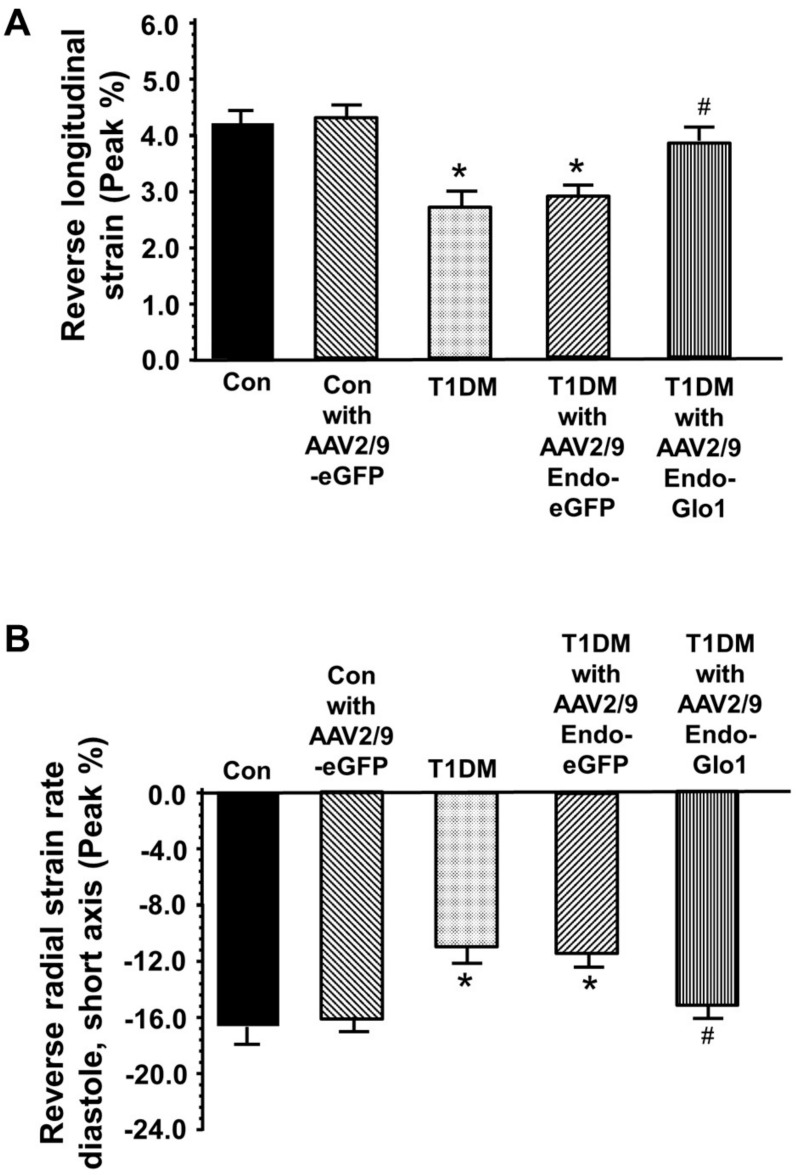
Bar charts showing reversed longitudinal (**A**) and radial strain rate (**B**) during early LV filling. For these measurements, the “reverse peak” algorithm in the Vevo Strain Software was utilized to analyze the B-mode images as described by Schnelle et al., 2018 [33]. Data shown are mean ± SEM. * denote significantly different from control (*p* < 0.05), # denote significantly different from T1DM (*p* < 0.05). The number of animals used per group is shown in the tables.

**Figure 3 antioxidants-09-00592-f003:**
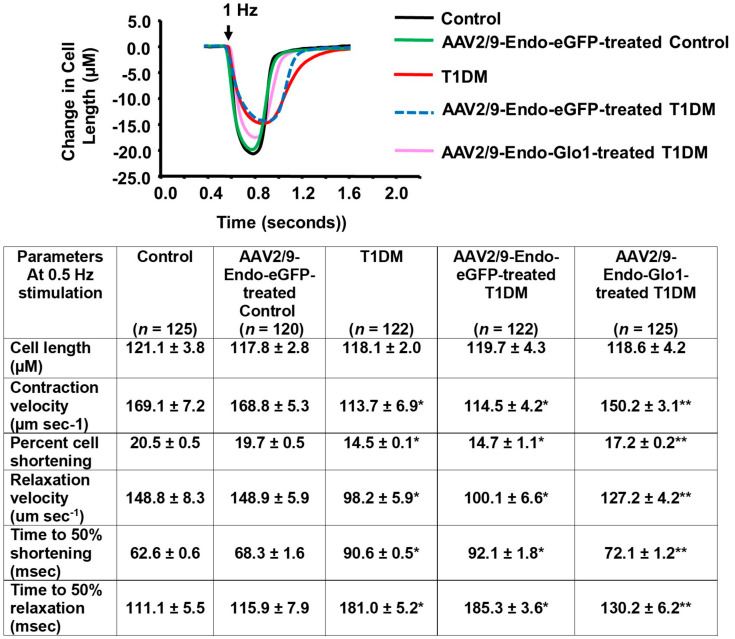
Data showing evoked contractile kinetics in left ventricular myocytes isolated from control, T1DM and T1DM-treated hearts. Images above are representative evoked contraction/relation profiles of ventricular myocyte isolated from the control, AAV2/9-Endo-eGFP-treated Con, T1DM, AAV2/9-Endo-eGFP-treated T1DM, and AAV2/9-Endo-Glo1-treated DM rats field-stimulated (10 V) for 10 ms at 0.5 Hz, using a pair of platinum wires placed on opposite sides of the chamber. The extent of myocyte shortening and rates of shortening and lengthening were determined using IonWizard Version 5.0 and means ± S.E.M for *n* > 120 cells from *n* = 5 animals are shown in the table below.

**Figure 4 antioxidants-09-00592-f004:**
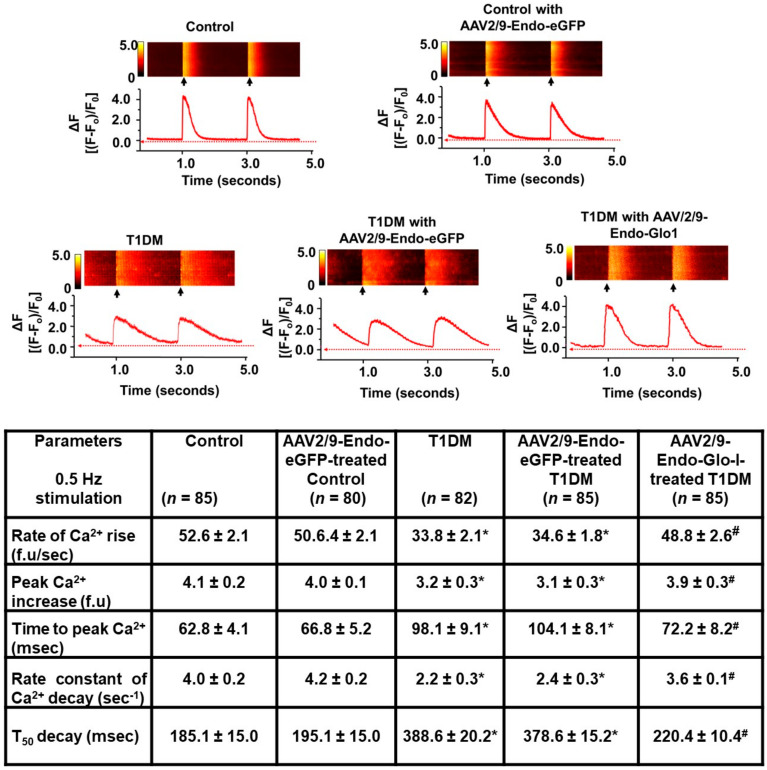
Original traces showing evoked Ca^2+^ transients in left ventricular myocytes isolated from the control, T1DM, and T1DM-treated hearts. Images above are representative line Ca^2+^ transients from ventricular myocytes isolated from the control, AAV2/9-Endo-eGFP-treated Con, T1DM, AAV2/9-Endo-eGFP-treated T1DM, and AAV2/9-Endo-Glo1-treated T1DM rats preloaded with the Ca2+ fluorophore dye Fluo-3 and field stimulated at 0.5 Hz. Data shown in Table below are mean ± S.E.M for *n* >80 cells from *n* = 5 animals are listed in the table below. * denote significantly different from the control (*p*< 0.05), # denote significantly different from T1DM (*p*< 0.05).

**Figure 5 antioxidants-09-00592-f005:**
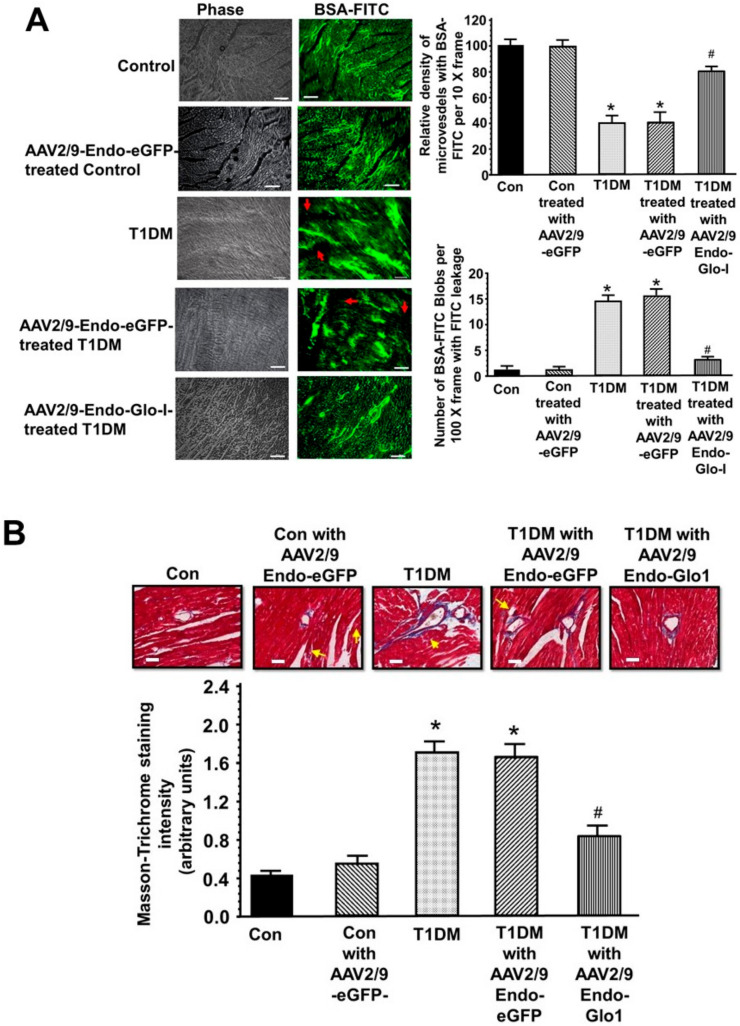
Vascular perfusion, permeability, and fibrosis in hearts from the control, T1DM, and T1DM-treated rats: (**A**) shows representative images of BSA-FITC in left ventricular tissues from the control, AAV2/9-Endo-eGFP treated control, T1DM, AAV2/9-Endo-eGFP-treated T1DM, and AAV2/9-Endo-Glo1-treated T1DM rats. Graphs on the upper right show the relative density of BSA-FITC per 10× frame and in the lower left, the relative density of BSA-FITC blobs per 10× frame (an index of vascular leakage). Values are mean ± S.E.M from *n* > 20 sections from *n* = 5–6 rats. White bar at bottom of each image = 50 µm. (**B**), shows representative 10× sections showing Masson Trichrome staining in left ventricular sections (apex) from control, AAV2/9-Endo-eGFP-treated Con, T1DM, AAV2/9-Endo-eGFP-treated T1DM and AAV2/9-Endo-Glo1-treated DM rats. Graph below is mean ± S.E.M from *n* > 20 sections from *n* = 4–5 male rats. * denote significantly different from the control (*p* < 0.05), # denote significantly different from T1DM (*p* < 0.05). White bar at bottom of each image = 50 µm.

**Figure 6 antioxidants-09-00592-f006:**
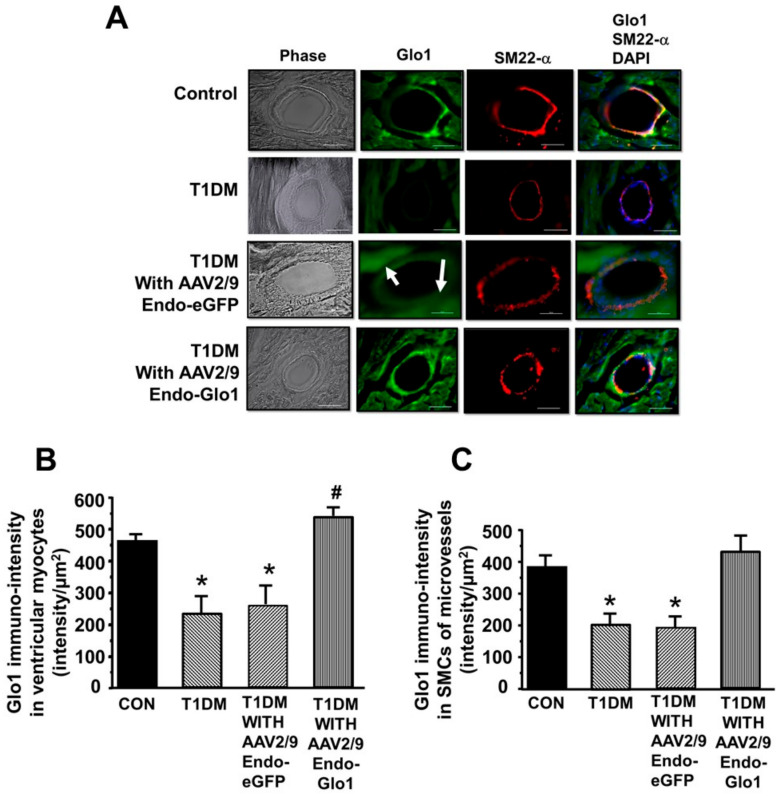
Glo1 levels in ventricular tissues. (A) shows immuno-fluorescence staining for Glo1 in myocytes and cSMCs from the control, T1DM, AAV2/9-Endo-eGFP-treated T1DM, and AAV2/9-Endo-Glo1-treated DM rats. White arrow emphasizes the diffuse staining associated with eGFP expression. Graphs in (**B**) and (**C**) are mean ± S.E.M. for *n* ≥ 30 sections obtained from *N* = 4–6 animals per group. (**D**) shows an autoradiogram for Glo1 protein in hearts from the control, AAV2/9-Endo-eGFP treated control, T1DM, AAV2/9-Endo-eGFP-treated T1DM, and AAV2/9-Endo-Glo1-treated T1DM rats. Graph below are mean ± S.E.M from ventricular homogenates from *n* = 5 rats done in duplicates. * denote significantly different from the control (*p*< 0.05), # denote significantly different from T1DM (*p* < 0.05). Bar at bottom of each image = 50 µm.

**Figure 7 antioxidants-09-00592-f007:**
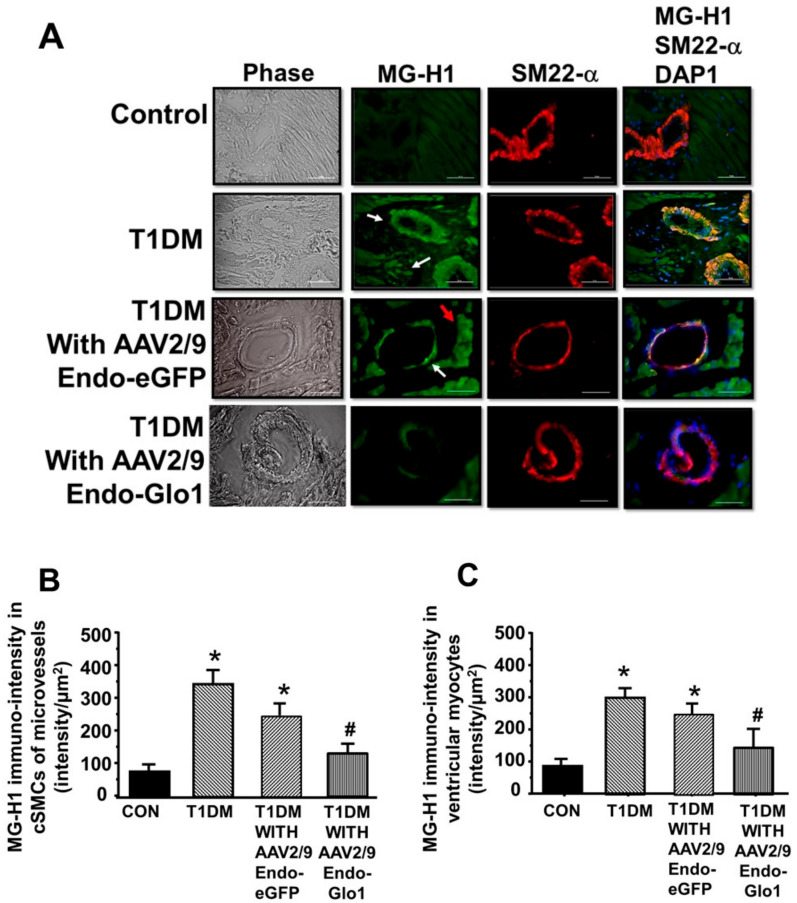
MG-H1 levels in ventricular tissues. (**A**) shows images of immuno-fluorescence staining for MG-H1 in cardiac myocytes and cSMCs from the control, T1DM, AAV2/9-Endo-eGFP-treated T1DM, and AAV2/9-Endo-Glo1-treated DM rats. Red arrow emphasizes the diffuse staining associated with eGFP expression. Graphs below in (**B**) and (**C**) are mean ± S.E.M. for *n* ≥ 30- sections obtained from *n* = 4–6 animals per group. (**D**) shows Coomassie-stained SDS-PAGE cell for the autoradiogram for MG-H1 in ventricular homogenates (**E**) from the control, AAV2/9- T1DM, AAV2/9-Endo-eGFP-treated T1DM, and AAV2/9-Endo-Glo1-treated T1DM rats. Bar chart graph in Panel (**F**) is mean ± S.E.M from ventricular homogenates from *n* = 3–5 rats done in duplicates. * denote significantly different from the control (*p* < 0.05), # denote significantly different from T1DM (*p* < 0.05). Bar at bottom of each image = 50 µm.

**Figure 8 antioxidants-09-00592-f008:**
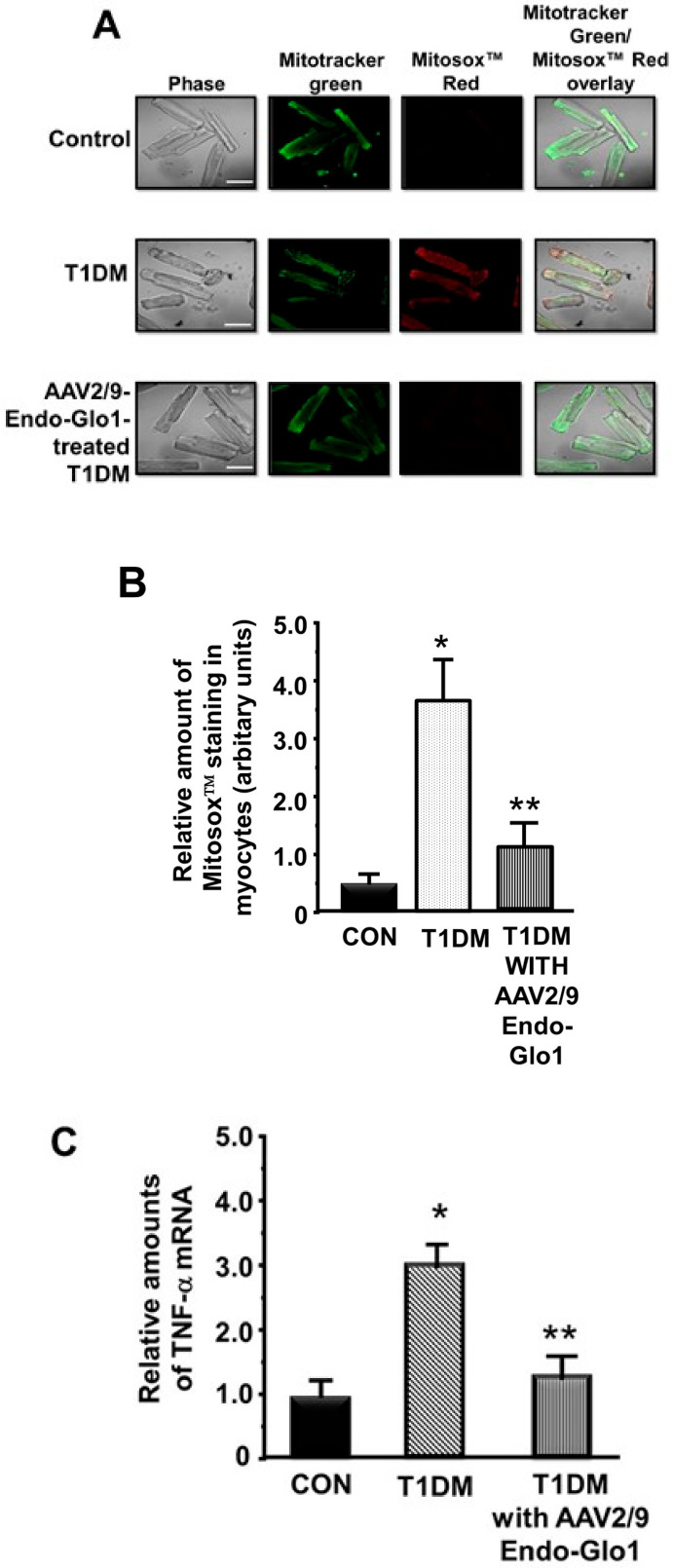
ROS and inflammation in ventricular myocytes. (**A**) shows representative Mitotracker Green (mitochondria) and MitoSox™ (ROS) fluorescence images in freshly isolated myocytes from the control, T1DM, and AAV2/9-Endo-Glo1-treated T1DM rat hearts. Graph below (**B**) is the mean ± S.E.M from >30 myocytes obtained from *n* = 5 animals. White bar = 50 µm. (**C**) shows TNF-α mRNA levels in left ventricular tissues from the control, T1DM, and AAV2/9-Endo-Glo1-treated DM normalized to β-actin. Values are mean ± S.E.M from *n* = 5 rats per group. (**D**) shows a representative autoradiogram for phosphorylated and non-phosphorylated p65 in left ventricular lysates from the control, AAV2/9-Endo-eGFP-treated control, T1DM, AAV2/9-Endo-eGFP-treated T1DM, and AAV2/9-Endo-Glo1-treated DM rats. Values shown in graph below are mean ± S.E.M from *n* = 3 rats per group. * denote significantly different from the control (*p* < 0.05), # denote significantly different from T1DM (*p* < 0.05).

**Table 1 antioxidants-09-00592-t001:** General characteristics of animals used in the study.

Measured Parameters	Control (*n* = 14)	Control Treated withAAV2/9- Endo-eGFP (*n* = 14)	T1DM (*n* = 20)	T1DM with AAV2/9-Endo-eGFP (*n* = 20)	T1DM with AAV2/9-Endo-Glo1 (*n* = 20)
Body mass (g)	389.2 ± 10.2	392.3 ± 11.4	275.4 ± 15.2 *	290.4 ± 12.3 *	320.3 ± 10.3 ^#^
Heart weight	1.2 ± 0.2	1.2 ± 0.2	1.0 ± 0.3	1.0 ± 0.3	1.1 ± 0.2
Heart to body weight ratio (mg/g)	3.1 ± 0.2	3.1 ± 0.2	3.6 ± 0.2 *	3.4 ± 0.2	3.4 ± 0.2
Blood glucose (mmol)	5.5 ± 1.6	6.3 ± 1.7	23.6 ± 4.5 *	24.1 ± 3.5	22.9 ± 3.2
% Glycated hemoglobin	4.2 ± 0.3	4.3 ± 0.4	7.6 ± 0.2 *	7.5 ± 0.3 *	7.2 ± 0.5
Serum insulin (ng/mL)	0.9 ± 0.2	1.0 ± 0.1	0.3 ± 0.1 *	0.3 ± 0.1 *	0.6 ± 0.1 ^#^
Serum TBARS activity (nmol/mL)	2.4 ± 0.2	2.3 ± 0.03	10.9 ± 1.1 *	11.1 ± 0.8 *	8.4 ± 1.2
Ventricular TBARS (nmol/mL)	2.1 ± 0.3	2.3 ± 0.3	8.2 ± 1.2 *	8.5 ± 1.0 *	5.2 ± 0.8 ^#^
Serum MG (µM)	0.3 ± 0.1	0.3 ± 0.1	1.2 ± 0.2 *	1.4 ± 0.2 *	1.2 ± 0.3
Ventricular MG (nmol/g)	2.0 ± 0.3	2.0 ± 0.5	8.9 ± 1.7 *	7.9 ± 2.1 *	3.8 ± 0.3 ^#^
Glo1 activity (µmol/min/100 mg ventricular tissue)	10.2 ± 1.0	11.4 ± 1.0	5.1 ± 0.8 *	6.3 ± 0.6 *	13.2 ± 1.2 ^#^

Values shown are mean ± S.E.M. for *n* = 18–20 animals per group. * denotes significant difference from Con (*p* < 0.05), # denotes significant difference from T1DM (*p* < 0.05).

**Table 2 antioxidants-09-00592-t002:** GSH:GSSG ratios and activities of γ-glutamylcysteine ligase and glutathione reductase.

Parameters Measured	Control (*n* = 6)	T1DM (*n* = 6)	AAV2/9-Endo-eGFP-Treated T1DM (*n* = 6)	AAV2/9- Endo-Glo-I-Treated T1DM (*n* = 6)
**GSH:GSSG ratio**	2.4 ± 0.2	1.55 ± 0.1 *	1.6 ± 0.3 *	2.3 ± 0.3 ^#^
**γ-glutamylcysteine ligase activity** **(mU/mg protein)**	470.5 ± 18.1	205.2 ± 1.2 *	210.5 ± 21.2 *	460.2 ± 20.4 ^#^
**glutathione reductase activity** **(mU/mg protein)**	48.1 ± 7.4	32.5 ± 8.0 *	30.5 ± 11.2 *	45.5 ± 10.1 ^#^

Values shown are mean ± S.E.M. for *n* = 4–6 animals per group. * denotes significant difference from Con (*p* < 0.05), # denotes significant difference from T1DM (*p* < 0.05).

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
