# Peer review of "Adeno-Associated Viral Transfer of Glyoxalase-1 Blunts Carbonyl and Oxidative Stresses in Hearts of Type 1 Diabetic Rats"

_antioxidants, 2020, doi:10.3390/antiox9070592_

Round 1

Reviewer 1 Report

Alomar and colleagues describe the cardioprotective effect of glyoxalase-1 (Glo1) after in vivo transduction by adeno-associated virus (AAV) in streptozocin-induced T1DM rats.

The manuscript is well designed and written. The materials and methods are detailed and accurate (quite rare to find today). The study is interesting and the results provide quite novel insights in diabetic cardiomyopathy.

Nevertheless, some issues need to be addressed:

  • The primer list for TNFα and β-actin qPCR is missing.
  • 1: How do authors explain the reduction in serum TBARS activity in control rats injected with AAV2/9-endo-eGFP when compared to untreated control (0.3 vs 2.4)? The data if analyzed are statistically significant (p≤0.0001).
  • After approximatively three days streptozocin—treated rats start displaying blood glucose levels greater than 300 mg/dL, therefore even if authors did not find significant difference in cardiac parameters, very likely MG already started its deleterious effect at cellular level. Did authors consider treating the cells at the very beginning (after 3 days) to assess whether the detoxification enzyme not only blunt but alsoprevents chronic hyperglycemia-induced cardiac dysfunction? Authors should discuss it.
  • Figure 5A: I cannot find red arrows in the middle rows indicating “halo or blob effect” described in text (line 360-362).
  • Figure 6A: the reactivity of SM22α looks higher in T1DM than control which is the opposite of what described in the text (line 390-391).
  • Figure 7B the loading control β-actin has to been shown.
  • Figure 8A provides, with different technique, the same information as Figure 6A. Authors should consider of moving the panel 8A in figure 6A. In figure 8A it is unclear what VAP-1 is for.
  • For completeness, authors should include, at least in the supplementary, the results for immunohistochemistry (Figure 6A and 6B), oxidative and carbonyl stresses, inflammation (Figure 7A and Figure 8) of the other control groups: control with AAV2/9-eGFP and T1DM with AAV2/9 endo-eGFP.
  • Considering the great emphasis on GSH in the discussion, the data described in paragraph “cellular GSH and GSH regulatory enzymes levels” should be displayed at least in a table.

Author Response

Reviewer #1

The manuscript is well designed and written. The materials and methods are detailed and accurate (quite rare to find today). The study is interesting, and the results provide quite novel insights in diabetic cardiomyopathy.

Response: Thank you

Concern #1: The primer list for TNFα and β-actin qPCR is missing. 

Response #1. This was included in the Antibodies and Reagent Section, 2.1. 

Comment #2: How do authors explain the reduction in serum TBARS activity in control rats injected with AAV2/9-endo-eGFP when compared to untreated control (0.3 vs 2.4)? The data if analyzed are statistically significant (p≤0.0001).

Response #2: This was an oversight, it should have been 2.3 not 0.3. 

Comment #3: After approximatively three days streptozocin-treated rats start displaying blood glucose levels greater than 300 mg/dL, therefore even if authors did not find significant difference in cardiac parameters, very likely MG already started its deleterious effect at cellular level. Did authors consider treating the cells at the very beginning (after 3 days) to assess whether the detoxification enzyme not only blunt but also prevents chronic hyperglycemia-induced cardiac dysfunction? Authors should discuss it

Response #3:  To date we have not treated animals after 3 days of T1DM to assess whether the detoxification enzyme can prevents chronic hyperglycemia-induced cardiac dysfunction. We have also not measured MG levels in hearts of rats three days after STZ injection.  In our hands we do not see any measurable changes in in vivo cardiac function and cardiac Glo1 protein levels until 18 days after STZ injection.

Comment #4: Figure 5A: I cannot find red arrows in the middle rows indicating “halo or blob effect” described in text (line 360-362)

Response #4.  This has been corrected. Thank you

Comment #5: Figure 6A: the reactivity of SM22α looks higher in T1DM than control which is the opposite of what described in the text (line 390-391)

Response #5:  We have replaced the panels in this Figure with a new set of images that better show lower SM22α immunoreactivity in smooth muscle cells of T1DM rats. Thank you

Comment #6: Figure 7B the loading control β-actin has to been shown. 

Response #6. Since MG-H1 immuno-reacts with many proteins in ventricular homogenates, we typically do not strip the PDVF membranes and reprobe it with a second primary antibody, i.e., ß-actin.  However, we have decided to include the original Coomassie-stain gel  to verify similar protein loading per lane. Our lab uses a semi-dry transfer procedure for Western blot assays and Coomassie-stain all SDS gels after the transfer.

Comment #7: Figure 8A provides, with different technique, the same information as Figure 6A. Authors should consider of moving the panel 8A in figure 6A. In figure 8A it is unclear what VAP-1 is for

Response #7: Thank you.  Figure 8A is now 6D. VAP-1 has also been removed. We regret this oversight.  We have also replaced the initial autoradiogram with one that shows multiple animals per group in a random manner, with β-actin as the internal reference.

Comment #8: For completeness, authors should include, at least in the supplementary, the results for immuno-histochemistry (Figure 6A and 6B), oxidative and carbonyl stresses, inflammation (Figure 7A and Figure 8) of the other control groups: control with AAV2/9-Endo-eGFP and T1DM with AAV2/9 Endo-eGFP

Response #8: The revised Figures now include data from all the groups that we have analyzed.  Unfortunately, we do not have data for all the groups for all assays due to resource limitations.

Comment #9: Considering the great emphasis on GSH in the discussion, the data described in paragraph “cellular GSH and GSH regulatory enzymes levels” should be displayed at least in a table

Response #9: Done.  It is now in Table 2.  Thank you

Reviewer 2 Report

Alomar et al report a study of AAV9 mediated glyoxalase-1 gene therapy for treating diabetic cardiomyopathy. They report effects of gene therapy on cardiac function, calcium cycling, remodeling, inflammation and oxidative stress in a rat model of streptozocin induced DM model. The study included sufficient number of animals and the analyses on cardiac pathology are extensive. The results are overall consistent, supporting the benefit of Glo-1 gene therapy. I have following comments.

Major:

Glo-1 expression is only examined by IHC and it lacks quantitative data. Please include viral genomic DNA and Glo-1 mRNA expression data to see the degree of over-expression. Since the authors used Endothelin promoter, I believe this is the optimal way to evaluate the level of over expression. Please also include GFP quantitation in controls. Related to this, the rationale for using endothelin promoter may be briefly mentioned in the introduction.

In Table 1, authors report improved animal weight. It is not clear how cardiac Glo-1 expression can increase body weight. Although the results show worse diastolic function, stroke volume is maintained. Thus the decrease in body weight is likely associated with cause other than cardiac pathology. It is possible that Glo-1 was expressed in other organs (such as skeletal muscle) and improvement in systemic condition improved cardiac function. Therefore, I suggest including data on Glo-1 expression in off-target major organs including SKM.

Minor:

The lower heart rate in the control groups (Table1) is a concern. It is likely due to deeper anesthesia and isoflurane can cause cardiac dysfunction (it could be that they are weak to anesthetic agents, but the results are still influenced). Please comment.

There are many typos and incomplete sentences. Please brush-up.

Author Response

Alomar et al report a study of AAV9 mediated glyoxalase-1 gene therapy for treating diabetic cardiomyopathy. They report effects of gene therapy on cardiac function, calcium cycling, remodeling, inflammation and oxidative stress in a rat model of streptozocin induced DM model. The study included sufficient number of animals and the analyses on cardiac pathology are extensive. The results are overall consistent, supporting the benefit of Glo-1 gene therapy.

Response: Thank you.

I have following comments

Comment #1: Glo-1 expression is only examined by IHC and it lacks quantitative data. Please include viral genomic DNA and Glo-1 mRNA expression data to see the degree of over-expression. Since the authors used endothelin promoter, I believe this is the optimal way to evaluate the level of over expression.

Response #1: We have measured Glo1 protein using Western blots (originally Figure 8A, now Figure 6D) and Glo1 activity (Table 1) in ventricular homogenates.  Unfortunately, at this time we do not have frozen tissues left for isolation of DNA/RNA to conduct the experiments suggested.  However, we do have a Western blot with multiple animals from the five groups and we have replaced the older autoradiogram in Fig 8A with the new one.  The samples were also loaded randomly on the gel to minimize bias.  In these studies, Glo1 expression in AAV2/9-treated T1DM rats was 25-30% higher in than that in control animals.  Thus, the virus we are using is expressing Glo1 to levels similar to that in the control, non-diabetic animals.  In future studies we will measure viral genomic DNA and Glo1 mRNA to more accurately define the degree of Glo1 expression with our virus.   Thank you.

Comment # 2: Please also include GFP quantitation in controls. Related to this, the rationale for using endothelin promoter may be briefly mentioned in the introduction

Response #2: We have included all the eGFP data that we have at this time in immunohistochemical and Western blot studies.  However, not all eGFP treated controls were analyzed due to a dwindling research budget. We have also included the rationale for using the endothelin-1 promoter in the last paragraph of the introduction.  It reads “Since inflammation is upregulated during T1DM, we created an AAV serotype 2 with capsid 9 driven by the promoter of the inflammation-regulated protein, endothelin-1 (AAV2/9-Endo-Glo1)” so as to express Glo1 under inflammatory conditions.

Comment #3: In Table 1, authors report improved animal weight. It is not clear how cardiac Glo-1 expression can increase body weight. Although the results show worse diastolic function, stroke volume is maintained. Thus the decrease in body weight is likely associated with cause other than cardiac pathology. It is possible that Glo-1 was expressed in other organs (such as skeletal muscle) and improvement in systemic condition improved cardiac function. Therefore, I suggest including data on Glo-1 expression in off-target major organs including SKM

Response #3:  We do not think AAV2/9-Endo-Glo1 treatment is increasing body mass.  Rather, is blunting the loss of body mass, but the specific mechanisms involved are not clear.  Since AAV2/9-Endo-Glo1 was administered intravenously, we expect effects some effects on end organs.  Thus far we have shown that AAV2/9-Endo-Glo1 administered intravenously prevents microvascular leakage in the brain (British Journal of Pharmacology (2016) 173 3307–3326, ref 24) and a manuscript is in preparation showing that it prevents glomerular hyperfiltration and microvascular leakage in the kidneys.  To date, we do not have any data on AAV2/9-Endo-Glo1 effect on SKM from these animals.  We have recently obtained IACUC approval to inject AAV2/9-Endo-Glo1 directly into the heart of T1DM rats to reduce off target effects.  Those data should be ready for publication in the 9-12 months.

Comment #4: The lower heart rate in the control groups (Table1) is a concern. It is likely due to deeper anesthesia and isoflurane can cause cardiac dysfunction (it could be that they are weak to anesthetic agents, but the results are still influenced). Please comment.

Response #4:  Several studies have shown that anesthesia can impair myocardial function, especially at higher doses. We typically induce anesthesia in rats by placing them in a chamber and exposing to 3% isoflurane for 3-4 minutes. Isoflurane (1-2%) is also used for maintenance of anesthesia for echocardiographic studies which last ∼ 10-12 minutes to minimize anesthesia-induced effects.  Stein et al., (Basic Res Cardiol 102:28–41 (2007)) earlier reported that low doses of isoflurane (1.5%) affords echocardiographic LV structural and functional data similar to those obtained in conscious rats. We have also revisited and corrected an error we found in the Table after you pointed this out.  Thank you.

Comment #5: There are many typos and incomplete sentences. Please brush-up

Response #5: Corrected. Thank you.

Reviewer 3 Report

In this paper, Alomar and collaborators show that preventing the accumulation of methylglyoxal in the heart of diabetic rats leads to stopping the development of diabetic cardiomyopathy. The concept of the study is elegant and the data are informative. However, there are some issues to be addressed:

  1. Page 1, lines 34-39: The phrases “Accumulation of methylglyoxal (MG) is an underlying cause of diabetic cardiomyopathy 24 (DC). Herein we investigated if expressing the MG-degrading enzyme glyoxalase-1 (Glo1) in rat 25 hearts shortly after onset of Type 1 diabetes mellitus (T1DM) would blunt DC development.” are redundant. They say mostly the same thing and they are not easy to follow. Please rephrase.
  2. Page 6, lines 275-277: “Administration of AAV2/9-Endo-Glo1 one week after STZ injection, blunted reductions in body mass but not glucose, insulin or % glycated hemoglobin levels. ” In table 1, the value of serum insulin is higher in AAV2/9-Endo-Glo1 than in T1DM group. This information is not found in the phrase above.
  3. Figure 5A: the pictures T1DM condition are not the same magnification as the others and it is difficult to compare images. Since the augmentation is higher, one cannot say that the same dark regions are not present in the other conditions. Moreover, these regions shown in the picture are small and they do not seem to indicate “decreased micro vessel perfusion and/or micro-ischemia”, but more likely are due to variations in tissue autofluorescence or camera exposure. Evaluation of microvessels density would have been better on transversal sections and using counterstaining with an endothelial marker such as CD31.
  4. Page 13, line 371, Figure 5 caption mentions an 20x magnitude, while in the figure and text, 10x can be found.
  5. Page 14, Figure 6. The pictures do not match the graphs. They are inversed. Please replace pictures with unaltered versions. The intensity of the fluorescence of the nuclei is not the same in all samples. 
  6. Page 17. Figure 8A. the graph shows the quantification of VAP-1, but the blot picture is not shown and VAP-1 evaluation is not mentioned in the text.
  7. The are typing errors throughout the manuscript. Some phrases are missing the verb (example, page 11 line 356 “No leakage of the BSA-FITC from the vasculature into the myocardium.”

Author Response

In this paper, Alomar and collaborators show that preventing the accumulation of methylglyoxal in the heart of diabetic rats leads to stopping the development of diabetic cardiomyopathy. The concept of the study is elegant, and the data are informative.

Response:  Thank you

However, there are some issues to be addressed

Comment #1: Page 1, lines 34-39: The phrases “Accumulation of methylglyoxal (MG) is an underlying cause of diabetic cardiomyopathy (DC). Herein we investigated if expressing the MG-degrading enzyme glyoxalase-1 (Glo1) in rat hearts shortly after onset of Type 1 diabetes mellitus (T1DM) would blunt DC development.” are redundant. They say mostly the same thing and they are not easy to follow. Please rephrase

Response #1: Done. It now reads “Accumulation of methylglyoxal (MG) arising from downregulation of its primary degrading enzyme glyoxalase-1 (Glo1) is an underlying cause of diabetic cardiomyopathy (DC).  Here we investigated if expressing Glo1 in rat hearts after the onset of Type 1 diabetes mellitus (T1DM) would blunt the development of DC”

Comment #2: Page 6, lines 275-277: “Administration of AAV2/9-Endo-Glo1 one week after STZ injection, blunted reductions in body mass but not glucose, insulin or % glycated hemoglobin levels.” In Table 1, the value of serum insulin is higher in AAV2/9-Endo-Glo1 than in T1DM group. This information is not found in the phrase above

Response #2: This has been corrected.  It now reads “Administration of AAV2/9-Endo-Glo1 one week after STZ injection, blunted reductions in body mass but not glucose or % glycated hemoglobin levels.”  There was a trend for lower glucose and % glycated hemoglobin in AAV2/9-Endo-Glo1-treated T1DM animals but the values did not attain statistical significance from T1DM animals.

Comment #3: Figure 5A: the pictures T1DM condition are not the same magnification as the others and it is difficult to compare images. Since the augmentation is higher, one cannot say that the same dark regions are not present in the other conditions. Moreover, these regions shown in the picture are small and they do not seem to indicate “decreased micro vessel perfusion and/or micro-ischemia”, but more likely are due to variations in tissue autofluorescence or camera exposure. Evaluation of microvessels density would have been better on transversal sections and using counterstaining with an endothelial marker such as CD31

Response #3:  The pictures in T1DM condition were taken the same magnification, but we tried to expand select areas for emphasis on the right panels. During acquisition and analyses of data, we also ensured that camera and intensity settings for all images from all groups were the same to avoid bias.  We have revised the labeling of the Y-axis of the upper graph of Fig. 5A to indicate “relative density of vessels with BSA-FITC inside per 10X frame” and the lower graph to indicate “number of BSA-FITC blobs per 10X frame.”

We have revised Figure 5A to to include AAV2/9-Endo-eGFP-treated control and AAV2/9-Endo-eGFP-treated  T1DM. The green fluorescence in the images in Fig 5A are not from tissue autofluorescence, instead they are from the BSA-FITC dye that we injected IV and allowed to circulate five minutes prior to sacrificing animals, removing and fixing the tissues.

We agree that higher magnification images are needed to define ischemia and this point has been dampened in the revised manuscript. In future studies we will counterstained with the endothelial marker CD31 (or eNOS) to more accurately define the density of microvessels.

Comment #4: Page 13, line 371, Figure 5 caption mentions an 20x magnitude, while in the figure and text, 10x can be found

Response #4: Corrected.  It should be 10X frame

Comment #5: Page 14, Figure 6. The pictures do not match the graphs. They are inversed. Please replace pictures with unaltered versions. The intensity of the fluorescence of the nuclei is not the same in all samples

Response #5: Corrected.

Comment #6: Page 17. Figure 8A. the graph shows the quantification of VAP-1, but the blot picture is not show and VAP-1 evaluation is not mentioned in the text.

Response #5: VAP-1 has been removed. Thank you.

Comment #6: The are typing errors throughout the manuscript. Some phrases are missing the verb (example, page 11 line 356 “No leakage of the BSA-FITC from the vasculature into the myocardium was observed. 

Response #6: Corrected.  Thank you.

Round 2

Reviewer 1 Report

All concerns have been addressed. 

Author Response

No comments